# Sociodemographic Associations of Dementia Literacy in Older Australians

**Joyce Siette** [1,2,*] [ID] **and Laura Dodds** [1,2] [ID]

1 The MARCS Institute for Brain, Behaviour and Development, Western Sydney University, Westmead, NSW 2145, Australia

2 Australian Institute of Health Innovation, Macquarie University, Macquarie Park, NSW 2113, Australia

* Correspondence: joyce.siette@westernsydney.edu.au

**Abstract:** Recent levels of dementia literacy in older Australian adults remains relatively unexplored. Our purpose was to identify whether dementia literacy has changed in older Australians, sociodemographic characteristics associated with better literacy, and barriers to dementia risk reduction. A 32-item adapted British Social Attitudes Survey was administered to 834 community-dwelling older adults (mean age 73.3, SD = 6.0, range 65–94) on dementia awareness and knowledge of dementia risk and protective factors. Descriptive analyses, logistic, and multiple linear regressions were used to examine sociodemographic factors on dementia awareness and literacy. Most respondents (61%) were aware of the relationship between different lifestyle factors and dementia risk, with the majority reporting cognitive (85.0%) and physical inactivity (83.4%) as key risk factors. Few were able to identify less well-known factors (e.g., chronic kidney disease; 15.8%). Individuals with higher educational attainment were more likely to agree that dementia is modifiable (OR 1.228, 95% CI 1.02–1.47). Younger age (β = −0.089, 95% CI −0.736–−0.065, p = 0.019) was significantly associated with a higher number of correctly-identified dementia risk factors. Lack of knowledge was the key barrier to hindering dementia risk reduction. A tailored, evidence-informed, population-based lifespan approach targeting dementia literacy may help alleviate commonly reported barriers and support dementia risk reduction.

**Keywords:** aging; dementia awareness; dementia prevention; public health; risk reduction behavior; health promotion; risk factors; sociodemographic factors





## 1. Introduction

The increasing prevalence of dementia has stipulated a global health priority [1,2]. Internationally, approximately 50 million people are affected by dementia, with this figure projected to triple by 2050 [3]. Dementia is a progressive neurodegenerative condition that can be caused by a range of disorders that affect brain function resulting in changes in cognition and behavior [4]. There are multiple types of dementia, and the most common in Australia is Alzheimer's disease, with a prevalence rate of 50–75%. This is followed by vascular and frontotemporal dementia and dementia with Lewy bodies. Alzheimer's disease is characterized in the early stages by changes in short-term memory, depressive symptoms and apathy, and later stages of behavioral and psychological changes [5]. Indeed, recent studies have shown that early signs of impairment in emotion recognition, attention deficits, and motor control difficulties are a reflection of earlier cognitive decline and are common in patients with neurodegenerative diseases [6–8].

Although there is yet no available cure for dementia [9], research into the pathophysiology of Alzheimer's disease has identified certain biomarkers (e.g., central nervous system enzyme alterations, gut health indicators) that show promise in paving the way for personalized clinical management and treatment plans to address neurodegenerative disease progression [10]. Complementary research also indicates the pivotal role of modifiable

lifestyle factors (e.g., physical inactivity, poor mental health, presence of chronic health conditions such as diabetes and hypertension) on late-life dementia development and risk [11,12]. Thus, a concerted focus on adopting healthy brain lifestyles at a population level to target dementia prevention has emerged [1,13–17].

Improving dementia literacy is the necessary first step to supporting and driving dementia risk reduction [16]. However, population-based surveys highlight large variances in dementia literacy and awareness, with generally limited knowledge of dementia symptoms and the belief that the condition is modifiable [18–30], even in high-income countries. European surveys covering six European countries highlighted limited awareness of early dementia signs and available therapies for Alzheimer's disease [22]. Similar results have been observed in the UK [23] and the Netherlands [19], which found that adults lacked awareness of risk factors and their influence on the relationship between lifestyle and the onset of dementia.

Within Australia, around 72% of the general population believed that dementia risk reduction was possible [26]; however, this result was largely driven by adults. In an Australian pilot survey of 56 older participants, 80% reported prior knowledge of dementia risk factors signifying high preexisting awareness, but there were also descriptions of highly varying attitudes towards treatment [20].

Nevertheless, earlier studies examining dementia literacy tend to have low response rates (e.g., [26]) and small sample sizes, which hinders generalizability. Indeed, most of the studies surveyed adults from 18 years of age [26,27] with high levels of education (e.g., [19,26,27]), resulting in a possible overestimation of dementia literacy [19,26,27]. Furthermore, these studies often do not report on the barriers to adopting lifestyle behaviors that support dementia risk reduction. These limitations require consideration when developing future studies to gather a holistic understanding of the nature of dementia literacy that exists within the older Australian community.

In order to support future strategies for dementia prevention in older adults, the aim of this study was to identify (i) current levels of dementia literacy in older Australians; (ii) the association of dementia literacy with sociodemographic; and (iii) barriers to adopting lifestyles supporting dementia risk reduction. This insight will assist with tailoring future effective public health interventions targeting better dementia awareness and motivation to adopt healthy lifestyle patterns.

## 2. Materials and Methods

### 2.1. Design

The current study formed part of the broader Brain Bootcamp research project, which aimed to decrease dementia risk and increase awareness of associated risk factors amongst older adults [31]. A cross-sectional survey was delivered predominantly online, with postal options available upon request. Ethics approval was granted by the Macquarie University Human Ethics Committee (protocol 9174).

### 2.2. Study Population and Recruitment

Community-dwelling older adults living in New South Wales (NSW), Australia, were recruited via media (i.e., newsletters, flyers, radio), e-newsletters, and flyers at not-for-profit organizations, local councils, libraries, residences, clinical and non-clinical settings to ensure the advertisement scheme obtained a representative and diverse sample. Respondents were eligible if they were (i) 65 years of age or older, (ii) without confirmed or self-diagnosis of dementia or severe depression, (iii) residing in NSW, and (iv) able to provide informed consent. Respondents who had self-reported memory problems were eligible to participate.

### 2.3. Procedure

Between January and March 2021, respondents were able to complete a survey that assessed their knowledge of modifiable dementia risk factors and sociodemographic factors.

### 2.4. Measures

Sociodemographic: included age, gender, and education, whereby the education variable was divided into low, middle, or high categories. These represented 0–6 years of education (i.e., primary or low vocational education); 7–12 years of education (i.e., intermediate secondary education or intermediate vocational education or university); and >12 years of education (i.e., higher vocational education or university). To classify remoteness and socioeconomic status, the Accessibility/Remoteness Index of Australia (ARIA) [32] and the Index of Relative Socioeconomic Advantage and Disadvantage (IRSAD) [33] were used, which calculates locality (i.e., metropolitan vs. regional; which consisted of inner and outer regional, rural and remote areas) and socioeconomic status (i.e., described in quintiles where 1 is low and 5 is high) based on their location of residency.

General dementia literacy, awareness, and barriers: An adapted 32-item validated questionnaire measuring dementia awareness and literacy was administered to respondents. It consisted of a mixture of 10 items from the UK's British Social Attitudes (BSA) survey [34] and 22 items from the MijinBreincoach public health campaign [35]. The questionnaire asked respondents about their belief in the possibility of dementia being a modifiable condition (i.e., *dementia literacy*) and was asked to show how much they agreed with the statement "There is nothing anyone can do to reduce their risk of dementia" on a 5-point Likert scale. The questionnaire also captured general awareness of 12 modifiable dementia risk protective factors (e.g., hypertension, depression, mental activity) (i.e., *dementia awareness*) and barriers to adopting lifestyles that support dementia risk reduction (i.e., *barriers*) [36,37]. An example of a dementia awareness statement is "High blood pressure increases your chances of getting dementia," where respondents were asked to indicate how much they agree with the statement on a 5-point Likert scale ranging from "Strongly Agree" to "Strongly Disagree". Internal consistency of the survey ranged from good to excellent, with a reported Cronbach's alpha for the dementia awareness subscale at $\alpha = 0.82$.

### 2.5. Statistical Analysis

The sample size was based on the primary outcome of dementia knowledge [35]. An earlier study examined a middle-aged and older sample with the same instrument [35]. Using their scores as a reference for sample size calculation with a 95% confidence interval, a beta error of 10%, and an alpha of 0.05, the required sample size was calculated using this formula: $n = ((Z_{1-\alpha}/2 \times \sigma)/\delta))^2$. A sample size of 400 was required based on this calculation. Considering a 50% return rate in survey studies, 600 participants were needed in the study.

Data were assessed for normality with descriptive statistics computed and Chi-square analyses calculated to identify associations between dementia literacy and awareness. Inferential statistics of binary logistic regression were further performed to identify the associations of sociodemographic characteristics with dementia literacy (a dichotomous, categorical variable with yes or no responses to the statement that dementia is a modifiable condition). In addition, multiple regression analyses were used to investigate linear associations between dementia awareness as defined by the number of recognized modifiable dementia risk factors (a continuous variable from 0 to 12) with age, gender, socioeconomic status, locality, and education. Model assumptions for the regression analyses were tested with homoscedasticity by visual checks with scatterplots and assessing for multicollinearity issues (VIF < 2). Statistical significance was set to $p < 0.05$. Analyses were conducted using the Statistical Package for the Social Sciences software (SPSS V27, IBM, Armonk, NY, USA).

## 3. Results

A total of 857 respondents completed the survey (Table 1). On average, respondents were 73.4 years old (SD = 6.2, range = 65–94), mostly women (70%), and living in a metropolitan area (77.7%). A large proportion had attained graduate studies (44.6%). The

majority were born in English-speaking countries (84.1%), and close to half of the sample were of high socioeconomic status (49.4%).

**Table 1.** Summary of participant characteristics (*n* = 857).

| Characteristic | *n* (%) |
|---|---|
| Gender | |
| Female | 597 (70.0) |
| Male | 256 (30.0) |
| Age (Mean (SD), range) | 73.4 (6.2), 65–94 |
| 65–69 | 276 (32.2) |
| 70–79 | 444 (51.9) |
| 80+ | 136 (15.9) |
| Education | |
| Low | 293 (34.6) |
| Middle | 176 (20.8) |
| High | 377 (44.6) |
| Country of birth | |
| English-speaking country | 719 (84.1) |
| Non-English-speaking country | 136 (15.9) |
| Socioeconomic Status | |
| 1 (lowest) | 43 (5.4) |
| 2 | 125 (15.8) |
| 3 | 138 (17.4) |
| 4 | 95 (12.0) |
| 5 (highest) | 391 (49.4) |
| Locality | |
| Metropolitan | 627 (77.7) |
| Regional/Remote/Rural | 180 (22.3) |
| Interest in receiving information to improve your brain health | |
| Yes | 820 (98.3) |
| No | 11 (1.3) |

### 3.1. Dementia Literacy

Of the total sample, 572 (69.1%) respondents stated that dementia risk reduction is possible, demonstrating a high level of awareness of the association between brain health and lifestyle factors (Table 2).

**Table 2.** The proportion of participant responses to the dementia awareness questionnaire.

| Statement | Agree | Neutral | Disagree |
|---|---|---|---|
| There is nothing anyone can do to reduce their risk of dementia | 106 (12.8) | 150 (18.1) | 572 (69.1) |
| High blood pressure contributes to dementia risk | 313 (37.8) | 435 (52.5) | 81 (9.8) |
| Smoking increases your chances of getting dementia | 419 (50.7) | 320 (38.7) | 88 (10.6) |
| No or moderate alcohol use lowers your chances of getting dementia | 392 (47.6) | 302 (36.7) | 129 (15.7) |
| Regular physical activity lowers your chances of getting dementia | 689 (83.4) | 81 (9.8) | 56 (6.8) |
| Depression increases the chances of getting dementia | 456 (55.0) | 296 (32.7) | 77 (9.3) |
| Diabetes increases the chances of getting dementia | 292 (35.4) | 448 (54.2) | 86 (10.4) |
| A mentally active lifestyle lowers the chances of dementia | 701 (85.0) | 75 (9.1) | 49 (5.9) |
| Heart disease increases the chances of getting dementia | 246 (29.8) | 475 (57.5) | 105 (12.7) |
| Kidney disease increases the chances of getting dementia | 129 (15.5) | 576 (69.4) | 125 (15.1) |
| High cholesterol increases the chances of getting dementia | 256 (31.0) | 444 (53.7) | 127 (15.4) |
| A healthy diet lowers the chances of getting dementia | 592 (71.8) | 180 (21.8) | 53 (6.4) |

Chi-square analyses showed that individuals residing in metropolitan areas had significantly higher levels of agreement that dementia risk reduction was possible compared to individuals from rural or regional areas ($\chi2(1) = 4.8$, *p* = 0.028) (Table 3). Similarly,

respondents who had higher levels of education ($\chi$2(2) = 6.421, *p* = 0.04) and younger age (<79 years) ($\chi$2(3) = 9.14, *p* = 0.01), had significantly higher levels of agreement. There were no other significant demographic associations for gender ($\chi$2(2) = 0.859, *p* = 0.651), age (F (31,817) = 1.688, *p* = 0.168), socioeconomic status ($\chi$2(8) = 9.248, *p* = 0.322), and country of birth ($\chi$2(2) = 2.034, *p* = 0.362).

**Table 3.** Association of demographic variables with the belief that dementia is a modifiable condition.

| Characteristic | Agree | Neutral | Disagree | $\chi$2/F | *p*-Value * |
|---|---|---|---|---|---|
| Gender | | | | | |
| Female | 70 (11.8) | 113 (19.1) | 409 (69.1) | 0.859 | 0.651 |
| Male | 36 (14.1) | 47 (18.4) | 172 (67.5) | | |
| Age (Mean (SD), range) | 73.3 (6.4) | 73.8 (6.7) | 73.1 (5.8) | 1.688 | 0.168 |
| 65–69 | 38 (13.8) | 53 (19.2) | 185 (67.0) | 9.140 | **0.010** |
| 70–79 | 49 (11.0) | 75 (16.9) | 320 (72.1) | | |
| 80+ | 20 (15.4) | 32 (24.6) | 78 (60.0) | | |
| Education | | | | | |
| Low | 39 (13.5) | 62 (21.5) | 187 (64.9) | 6.421 | **0.040** |
| Middle | 25 (14.2) | 36 (20.5) | 115 (65.3) | | |
| High | 42 (11.2) | 60 (16.0) | 274 (72.9) | | |
| Country of birth | | | | | |
| English-speaking country | 86 (12.0) | 131 (18.3) | 497 (69.6) | 2.034 | 0.362 |
| Non-English-speaking country | 21 (15.6) | 28 (20.7) | 86 (63.7) | | |
| Socioeconomic Status | | | | | |
| 1 (lowest) | 4 (9.3) | 7 (16.3) | 32 (74.4) | 9.248 | 0.322 |
| 2 | 12 (9.6) | 25 (20.0) | 88 (70.4) | | |
| 3 | 10 (7.2) | 32 (23.2) | 96 (69.6) | | |
| 4 | 15 (15.8) | 18 (18.9) | 62 (65.3) | | |
| 5 (highest) | 56 (14.3) | 65 (16.6) | 270 (69.1) | | |
| Locality | | | | | |
| Metropolitan | 82 (13.2) | 120 (19.3) | 420 (67.5) | 4.800 | **0.028** |
| Regional/Remote/Rural | 16 (8.9) | 31 (17.3) | 132 (73.7) | | |

* Bold values in this column indicate a significant association.

Binary logistic regression analysis confirmed the association of education. Respondents with more than 12 years of schooling (OR 1.228, 95% CI 1.02–1.47, *p* = 0.028) were 2.4 times more likely to agree that dementia was modifiable (Table 4). The logistic regression model was statistically significant, $\chi$2(6) = 13.1, *p* = 0.042, explained 2.4% (Nagelerke $R^2$) and correctly classified 69.9% of cases.

**Table 4.** Predictors of dementia-related beliefs based on binary logistic regression (*n* = 786).

| Independent Variable | Odds Ratio | 95% CI | *p*-Value * |
|---|---|---|---|
| Gender (reference category: male) | 0.762 | 0.546–1.064 | 0.111 |
| Age (reference category: 65–69 years) | 0.955 | 0.752–1.213 | 0.705 |
| Education (reference category: <12 years) | 1.228 | 1.023–1.474 | **0.028** |
| Country of birth (reference category: non-English-speaking country) | 0.805 | 0.530–1.222 | 0.308 |
| Socioeconomic Status (reference category: lowest) | 0.996 | 0.996–0.857 | 0.960 |
| Locality (reference category: regional) | 0.123 | 1.478–0.899 | 0.123 |

CI = confidence interval. * Bold values in this column indicate a significant association.

### 3.2. Association of Identified Dementia Risk and Protective Factors with Demographic Variables

Nearly half of the total sample (48.7%) correctly identified more than six of the 12 factors, with 3.5% correctly identifying all factors and only 5.4% unable to identify any of the factors (Figure 1). In terms of identified risk and protective dementia factors amongst participants, a mentally active lifestyle was commonly reported (84.1%), followed by physical activity (82.6%), healthy diet (71.0%), and depression (54.7%). Vascular factors such as high

blood pressure (37.5%), hypercholesterolemia (30.7%), and coronary heart disease (29.5%) were less well recognized. Chronic kidney disease (15.5%) was the least identified factor (Figure 2).

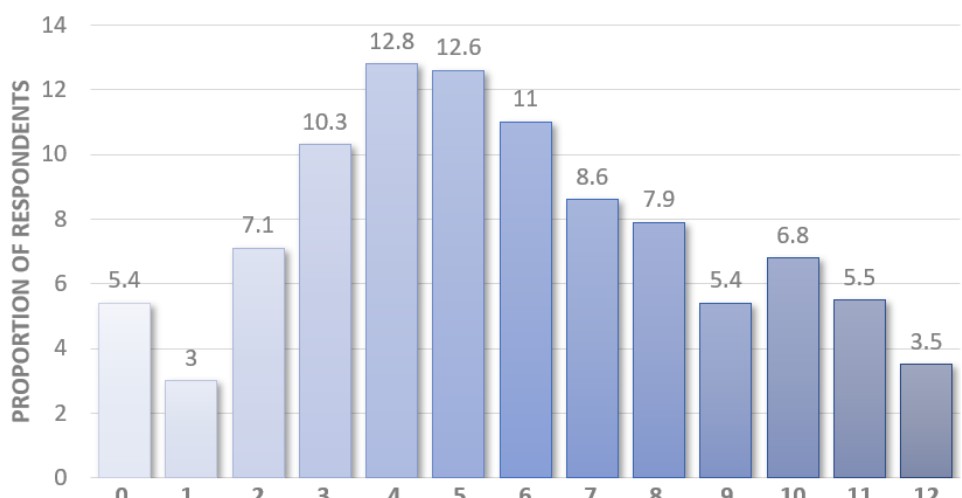

**Figure 1.** The proportion of respondents who correctly identified the number of risk and protective dementia factors (out of 12).

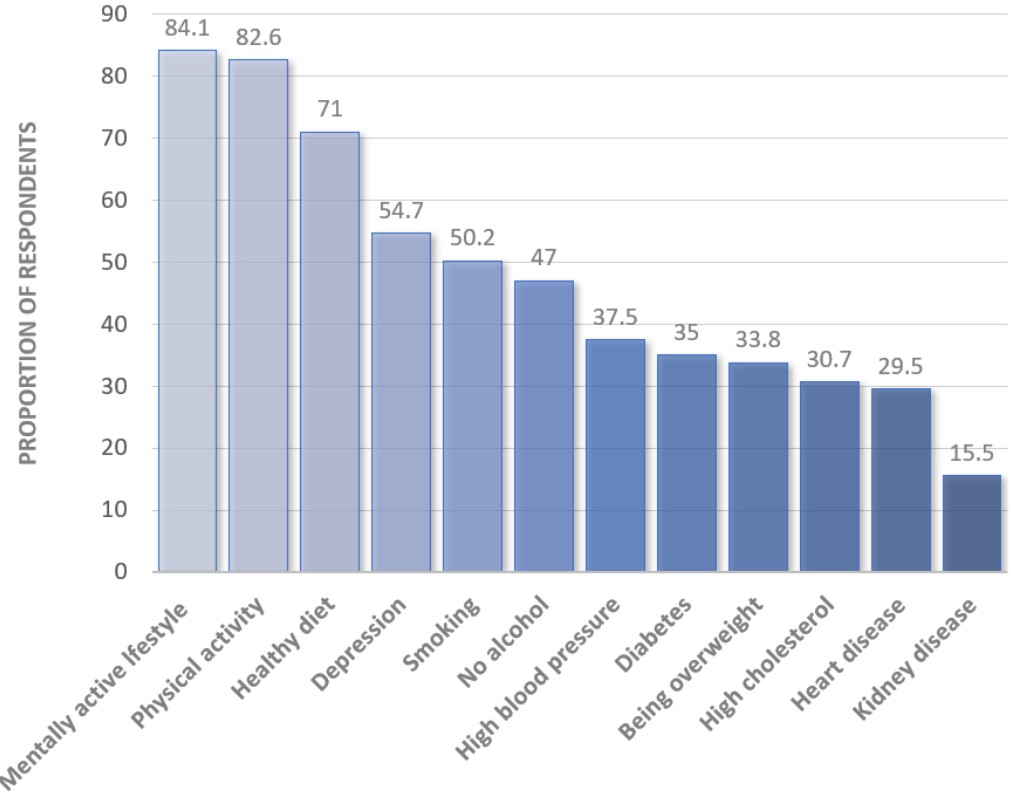

**Figure 2.** The proportion of respondents who correctly identified actual modifiable dementia risk factors.

Results from the multiple regression are shown in Table 5. When adjusting the regression analyses for the independent explanatory variables, significant associations remained only for age (F (6,757) = 2.122, *p* = 0.049). Adults aged between 65–69 years were significantly associated with being able to identify more modifiable dementia risk factors (β = −0.089, 95% CI −0.736–−0.065, *p* = 0.019). Tests to see if the data met the assumption

of collinearity indicated that multicollinearity was not a concern (Tolerance < 1, VIF < 2 for all variables).

**Table 5.** Multiple linear regression model of dementia literacy with sociodemographic variables (*n* = 764).

| Variable | B | SE | β | 95% CI | *p*-Value * |
|---|---|---|---|---|---|
| Constant | 4.896 | 1.026 | | 2.881–6.910 | |
| Gender | −0.201 | 0.242 | −0.03 | −0.675–0.273 | 0.406 |
| Age | −0.400 | 0.174 | −0.086 | −0.736−−0.065 | **0.019** |
| Education | 0.167 | 0.309 | 0.020 | −0.440–0.775 | 0.588 |
| Country of birth | 0.458 | 0.343 | 0.062 | −0.215–1.131 | 0.182 |
| Socioeconomic Status | 0.208 | 0.108 | 0.088 | −0.003–0.419 | 0.054 |
| Locality | 0.141 | 0.131 | 0.040 | −0.116–0.397 | 0.282 |
| | | | $R^2$ | 0.017 | |
| | | | F | 2.112 | 0.049 |

B = unstandardized beta; SE = standard error for the unstandardized beta; β = standardized coefficient beta; CI = confidence interval. * Bold values in this column indicate a significant association.

### 3.3. Barriers to Dementia Risk Reduction

Most respondents reported obtaining information about dementia from web searches (74.6%), their General Practitioner (52.2%) with some acquiring information from public sources, including not-for-profit organizations (e.g., Dementia Australia) (31.4%) and the library (15.5%; Figure 3). A minor proportion of respondents reported obtaining information from other sources (5.3%), and some reported not knowing where to obtain information (4.9%). Key barriers to dementia risk reduction included lack of knowledge (43.8%), lack of motivation (17.4%), lack of time (14.7%), and financial reasons (12.1%; Figure 4). Difficulty in organizing lifestyle change (9.1%), existing health problems (5.2%), and other barriers (2.4%) were less reported. A minority of respondents reported not knowing how barriers to dementia risk could be reduced (5.6%).

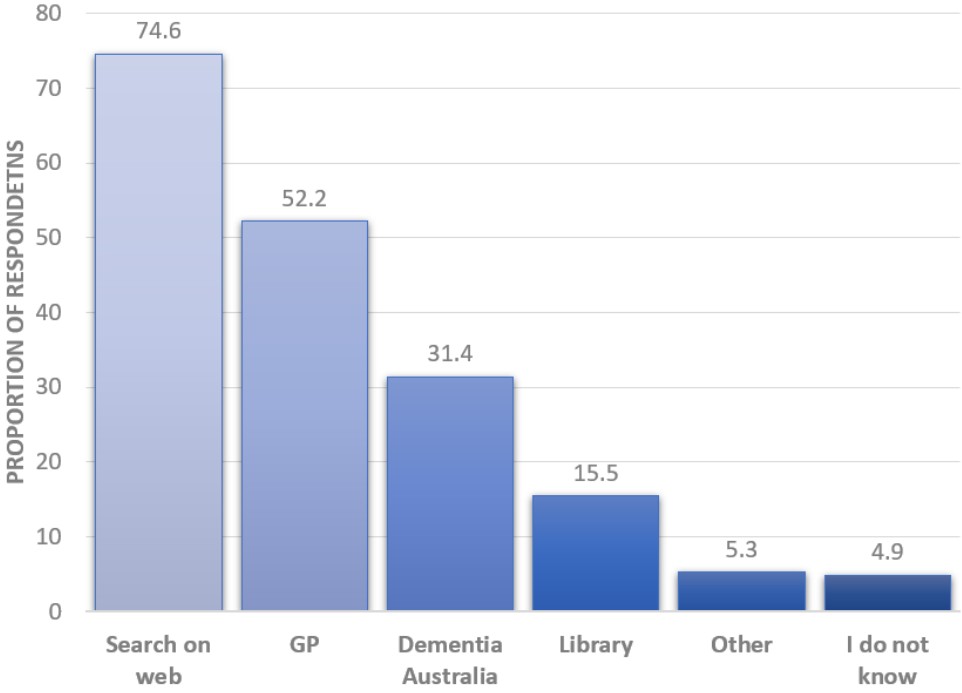

**Figure 3.** The proportion of respondents who reported on different information sources to support their dementia literacy.

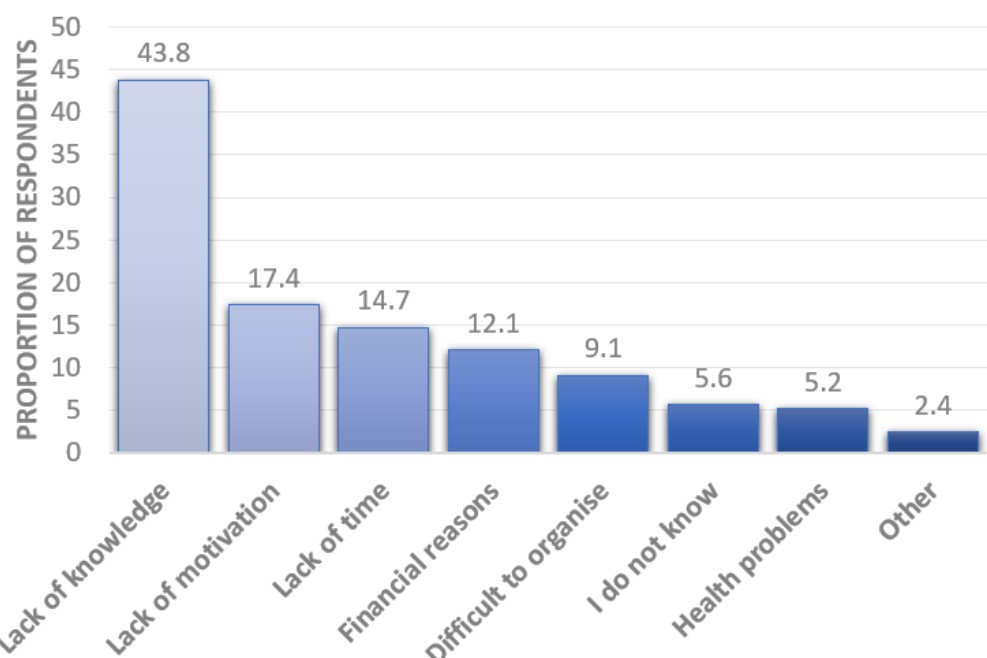

**Figure 4.** The proportion of respondents who reported barriers towards dementia risk reduction.

## 4. Discussion

The present study provides the latest snapshot of dementia literacy in a large, older Australian adult sample. The current cohort demonstrated advanced dementia literacy, with over three-quarters able to correctly identify more than four modifiable protective risk dementia factors. Whilst active cognitive lifestyles were the most recognized factor, awareness of cardiovascular risk-related factors remained relatively low. Education was the main driver of the knowledge that dementia is a modifiable condition, whilst younger age was associated with a higher number of correctly identified dementia risk factors. The key barrier to modifying lifestyle changes to support dementia risk reduction was a lack of knowledge. Our findings highlight the significance of educating the public and suggest the use of multi-level implementation actions and behaviors from clinicians, organizations, communities, and governments, to support dementia prevention at a population level.

### 4.1. Dementia Literacy

Relative to prior international research [15,19,22,23,28,30], our current findings on dementia literacy are considerably positive, with 69% of our respondents aware of the modifiable nature of dementia and only 5.2% unable to identify any risk factors. This is in contrast to international perspectives, where middle-aged and older adults had low levels of awareness. The general Dutch adult population sample found that 44% were aware of the relationship between brain health and lifestyle factors, with 10% unable to recognize any factors [34]. Van Asbroeck and colleagues [38] found that 34.5% of older Belgium adults were aware that dementia risk was modifiable through lifestyle changes, and nearly a fifth [21.6%] were unable to correctly identify a single risk factor. Similarly, just over half (51.5%) of American older adults believed in the possibility of reducing dementia risk [37]. In New Zealand, across a sample of 304 middle to older age adults (range 50 to 93), only four out of the fourteen modifiable risk or protective factors for dementia were correctly identified [39,40]. In America, from a sample of 703 participants of Chinese older adults, it was identified that, in fact, dementia knowledge decreased, and dementia beliefs remained unchanged from the period of 2013 to 2017 [41].

Australian-based studies revealed similar positive trends in dementia literacy compared to our present study. In 2009, Low and Anstey identified that 72% of community-dwelling adults had the opinion that dementia risk could be reduced [26]; and in 2015,

another Australian study found that just under half of the participants believed that dementia risk reduction was possible [27]. However, these results may be reflective of a younger sample in both studies (20 to 75 years of age) or increased awareness as a result of recent campaign strategies [27]. Furthermore, a recent systematic review collated findings from four surveys and showed that most Australians agreed that dementia risk reduction was possible (51%) or were unsure (28%), whilst the belief that dementia is an inevitable result of aging was estimated to be around one in five [42–45].

Dementia literacy in our current study thus appears to be more advanced compared to previous Australian older adult studies [25–27]. This may be partly explained by the sociodemographic characteristics of the current cohort, as most participants were well-educated, born in Australia, and around half were from the top socioeconomic quintile. Having more years of formal education was significantly associated with greater dementia awareness and better dementia literacy. Often the education an individual receives provides a foundation for seeking out novel learning experiences which may explain the higher levels of dementia literacy in our sample [43]. Indeed, Zhang and colleagues identified that education and other lifetime exposures to mental and social activities might provide greater awareness of the symptoms of mental diseases, such that individuals may have more access to greater resources that promote dementia literacy [30].

However, educational attainment across individuals and populations is dynamic and highly variable due to numerous factors, including but not limited to genetics, nutrition, health, parental socioeconomic status, and environmental and socioemotional influences [43]. Although our study does not take into account informal and other means of knowledge acquisition and development, taken together with previous research [30,43], it suggests that education, in its many forms, may be an avenue of intervention for bolstering dementia literacy for older adults.

In line with our findings, most international and national surveys found that having a cognitively active lifestyle was the most recognized lifestyle factor [24,27,34,38,42]. Physical activity and a healthy diet were among the top three factors most recognized [34,38,42], followed by mental stimulation and social activity [23,27]. Although these results are promising, a large number of significant, modifiable lifestyle factors are often overlooked [34]. Our study reaffirms that older adults are often unaware of the relationship between vascular issues (i.e., smoking, hypertension, high cholesterol) and brain health [27,34], as well as factors such as chronic kidney disease and coronary heart disease, which often remain unrecognized amongst older adult populations [38]. Here, the concern is that while awareness about the relationship between these lifestyle factors and dementia risk is low, vascular issues, chronic kidney disease, and coronary heart disease contribute to a significant proportion of dementia risk [44–47] and should be placed on the individual's radar.

In contrast to prior research, our study found minimal associations between sociodemographic characteristics and dementia literacy. Previous studies report that individuals who are male, have low levels of education, and are older, are less aware of the possibility of dementia risk reduction [34,48] and have poorer knowledge regarding the prevalence, symptoms, and treatment of dementia [30]. This discrepancy may be explained through a comparison of the sociodemographic features between the present sample and those in prior research. For instance, an earlier cross-country study reported on gender differences. However, most of its male participants were more highly educated than their female counterparts [47]. Moreover, a general difference is that research surrounding dementia literacy often captures a large age range (e.g., 20 to 70), whilst the present sample solely focused on the understanding of older adults only (range 65–94).

Younger age group membership (<79 years) was associated with better dementia literacy regarding the number of modifiable risk factors. One potential explanation may be considering factors commonly associated with aging and general health literacy, e.g., the decline in physical function, general loss of cognitive ability, and lower capacity for self-management, which prevents health-related information seeking and understanding [49]. However, only a small portion of our sample reported health problems as a barrier to

seeking information about dementia; thus, lower health literacy amongst older groups in our sample may be due to other factors. For example, lack of time due to caregiver responsibilities, the perception of being 'too old' to benefit from risk modifying information, and the fact that motivation to learn new things declines with age, alongside the confidence and the belief that we can use this information to change our circumstances [50]. Age is the most significant predictor of dementia [1]; thus, the association between dementia literacy and age has important help-seeking, screening, and diagnostic implications for dementia amongst the oldest adults [51]. Our results highlight that more concerted efforts need to be made to tailor salient educational actions toward later life stages. This includes utilizing primary health care clinicians, such as educating GPs, and nurses, to invest in enhancing dementia literacy which will ultimately be an invaluable asset for promoting healthy aging [49].

### 4.2. Barriers to Dementia Risk Literacy and Reduction

Barriers to adopting lifestyle changes to support dementia risk reduction were varied. Corresponding to prior studies, knowledge gaps were key contributors to less lifestyle behavior change [24,34]. Similarly, lacking the motivation to change was a prominent barrier to reducing dementia risk and can be partly explained by the socioemotional selectivity theory for behavior change, which states that older adults are less inclined to make future-orientated goals and tend to favor present-orientated goals [52]. Health problems have previously been identified as a major barrier and could be explained through the 'here-and-now' hypothesis for individual proneness to behavior change, which states that the greater the preoccupation with an existing condition, the less the time to adequately engage in behavior risk management for future health problems such as dementia [52]. Despite there being a small proportion reporting this as a barrier (5.2%), our findings included older adults who were unable to reduce their barriers to support dementia risk reduction. This suggests that there are individuals who may have pre-existing chronic health conditions that may be difficult to manage and, thus, are less likely to seek dementia-specific information.

We found that GPs were key supporters in assisting older adults' dementia risk literacy. Yet only 22% of GPs describe their dementia knowledge as adequate [53]. This reflects a potential shortfall within the primary care system to properly support patients' understanding and knowledge. Whilst family, caregivers, and religious leaders are often endorsed supporters by older adults [28], our study did not find these networks as an area of support. This discrepancy may be explained due to the previous study mainly targeting the low-to-middle-income demographic, whilst the current study has a broader demographic. This difference is particularly pronounced in countries with limited mental and general health resources that heavily rely on close and familiar networks that often become a common port of call for help-seeking and obtaining information pertaining to dementia [28].

### 4.3. Limitations

Our study's primary strength was adopting strategies to ensure a large sample size and adequate representation of the broader public. Despite our efforts, respondents were predominantly from high socioeconomic, English-speaking backgrounds with moderately high educational levels. Consequently, the results could be biased as the participants may be more knowledgeable and actively involved in the community and, therefore, may not accurately represent the general population. Furthermore, the study mainly recruited older adults who were eager to learn about dementia risk reduction and keen to change accordingly. Addressing such limitations in future studies provides an opportunity to gather evidence to inform effective intervention strategies that will propagate the benefits of dementia awareness and literacy for potentially more vulnerable populations in addition to healthier older adults [30,54–56].

*4.4. Implications*

Firstly, our study highlights that public health initiatives should be implemented to reduce dementia risk in older adults by better understanding the personal, environmental, cultural, and political determinants that influence dementia across the life course [57]. This will reduce confusion around risk factors and benefit healthy aging in general across populations. Secondly, future interventions should use a multifactorial, multi-level approach to public dementia education and risk reduction, whereby they aim to modify an individual's perceptions and beliefs around behavior change and risk reduction by integrating evidence-based strategies to tailor and deliver health information sessions. For the general public, these could commence in early to mid-life [24,36,58] and operate in conjunction with a long-term, concerted effort to target the systems that they interact with, including clinicians, organizations, and the government to support dementia prevention and the application of preventive interventions [12,13].

Thirdly, identifying patients at higher risk for dementia allows for the application of recent evidence and further research into novel clinical methods of Alzheimer's treatment and management techniques (e.g., non-invasive brain stimulation (NIBS) and invasive brain stimulation (IBS)). Despite being relatively in the early stages of development and application as treatments for Alzheimer's disease, recent evidence highlights the potential for enduring benefits, including the ability to excite critical brain regions involved in cognition and memory functions [59–61]. Concurrent with lifestyle modification and cognitive rehabilitation programs, these may provide a comprehensive and effective strategic direction for preventing the progression of dementia.

Finally, as our sample included an insufficient number of participants from culturally and linguistically diverse backgrounds, public health campaigns need to firstly establish awareness levels of these groups in their research and develop culturally appropriate and targeted messages and materials within their campaigns to increase population-wide dementia awareness [62–64].

## 5. Conclusions

According to the World Health Organization, healthy aging is considered the promotion and maintenance of functional capacity that allows well-being as we age and has recently been developed into a Healthy Ageing model. Dementia risk reduction actions should embody the key values of this approach amongst others, such as the Health Belief Model [65], so that brain healthier lifestyles can be readily accepted and adopted and health literate attitudes towards dementia treatment and management can be successfully implemented. This means understanding the relationship between intrinsic capacity (the physical and mental capabilities and motivations of the person or community) and environmental resources, including individual-level components (e.g., relationships and community participation) and contextual influences (e.g., service and treatment access, social policy and physical surroundings) [66]. Considering dementia and risk reduction as a holistic process and not limited to older age may extend to benefits in incidence and prevalence across populations over time [12]. Our study highlights that older adults have high dementia literacy, suggesting some success of recent public campaign interventions targeting dementia awareness and risk reduction. Incorporating evidence-based, multifaceted approaches that support education and awareness building amongst individuals, population sub-groups, and their environment is likely to eliminate key barriers and promote proactivity towards dementia risk reduction strategies for older adults in the future.

**Author Contributions:** Conceptualization, J.S. and L.D.; methodology, J.S.; formal analysis, J.S.; investigation, J.S.; resources, J.S.; data curation, L.D.; writing—original draft preparation, J.S.; writing—review and editing, L.D.; visualization, J.S. and L.D.; supervision, J.S.; project administration, J.S. and L.D.; funding acquisition, J.S. All authors have read and agreed to the published version of the manuscript.

**Funding:** This research was funded by the NSW Government, My Community Project grant number MCP19-04026.

**Institutional Review Board Statement:** The study was conducted in accordance with the Declaration of Helsinki and approved by the Ethics Committee of Macquarie University (protocol code 9174, 17 December 2020).

**Informed Consent Statement:** Informed consent was obtained from all subjects involved in the study.

**Data Availability Statement:** The data presented in this study are available on request from the corresponding author. The data are not publicly available due to restrictions because they contain information that could compromise the privacy of the research participants.

**Acknowledgments:** The authors would like to acknowledge Lakshmi Aravamudhan, Katie Leano, and Ilhias Pirzad for their contributions to the write-up and references of this manuscript.

**Conflicts of Interest:** The authors declare no conflict of interest. The funders had no role in the design of the study, in the collection, analyses, or interpretation of data, in the writing of the manuscript, or in the decision to publish the results.

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
