# Peer review of "Sociodemographic Associations of Dementia Literacy in Older Australians"

_2673-9259, doi:10.3390/jal2040021_

Round 1
Reviewer 1 Report
Thank you for giving me the opportunity to review this manuscript. The manuscript is well written, but I would like to suggest a few comments for authors consideration.
1. The authors used a 32-item validated scale to assess the main outcome measure. Since, this was the first study that uses this scale for your studied population, I would suggest the authors to briefly describe and analyze the psychometric properties of the scale for your studied population.
2. Indeed, please report the internal consistency of the scale.
3. It is unclear, how variable selection was done in the multivariate models? This needs to be explained.
4. The sample size calculation and formula used needs to be clearly described. What was the sampling technique used?
5. Please report your VIF values in your regression models.
Author Response
- The authors used a 32-item validated scale to assess the main outcome measure. Since, this was the first study that uses this scale for your studied population, I would suggest the authors to briefly describe and analyze the psychometric properties of the scale for your studied population.
Response: This is an excellent suggestion. The Cronbach alpha has been computed and added.
- Indeed, please report the internal consistency of the scale.
Response: See above.
- It is unclear, how variable selection was done in the multivariate models? This needs to be explained.
Response: Only demographic variables were captured in the survey and thus were selected for the models as they are considered related to the outcome in the literature. Our aims clearly state that we wish to explore how sociodemographic variables influence dementia literacy.
- The sample size calculation and formula used needs to be clearly described. What was the sampling technique used?
Response: A sample size calculated is now included. Representativeness of our participants as a cross-sectional survey was maintained by extending the population sample based on a convenience sampling procedure according to power analyses and sample size estimations as described below. The sample size was based on the primary outcome which was dementia knowledge. An earlier study examined a sample of a similar instrument. Using their scores as a reference for sample size calculation with a 95% confidence interval, a beta error of 10% and an alpha of 0.05, the required sample size was 388 and bumped to 776 considering a 50% return rate in survey studies.
- Please report your VIF values in your regression models.
Response: We have now included the VIF values in the regression models and statements about collinearity.
Reviewer 2 Report
Dear Authors,
Thank you for this interesting research, but there are many points that must be taken into account
- A literature review should be added to this paper, where it was found that the research does not contain a theoretical aspect or a scientific background except for the introduction.
-What is the criterion used to determine the sample size?
- Provide a justification for the lack of hypotheses for this study.
- Why didn't the researchers use multiple regression and did logistic regression? What is the reason for that?
-Justify the use of logistic regression analysis? What is the reason for that?
-The results obtained should be discussed in a more professional manner.
- Explain the theoretical implications of the study and reconsider the practical implications.
- Where are the determinants of future study and research?
Best regards
Author Response
- A literature review should be added to this paper, where it was found that the research does not contain a theoretical aspect or a scientific background except for the introduction.
Response: A literature review is provided in the introduction as well as how the results relate to the literature in the discussion. Additional background information has been added to these based on other reviewer comments and recommendations. We are unclear as to where we are supposed to add a supplementary literature review into the paper as it does not fit within the methods or results.
- What is the criterion used to determine the sample size?
Response: The sample size was based on the primary outcome which was dementia knowledge. An earlier study examined a sample of a similar instrument. Using their scores as a reference for sample size calculation with a 95% confidence interval, a beta error of 10% and an alpha of 0.05, the required sample size was 388 and bumped to 776 considering a 50% return rate in survey studies.
- Provide a justification for the lack of hypotheses for this study.
Response: Thank you for the suggestion. Hypotheses are not usual or standard for this field of public health research which is where this study is more aligned, thus a hypothesis has not been included.
- Why didn't the researchers use multiple regression and did logistic regression? What is the reason for that?
Response: Thank you for the opportunity to clarify. Both logistic and multiple linear regression were conducted. For the belief on whether dementia was modifiable (or not), this was coded as a dichotomous, categorical variable so logistic regression was chosen. The second dependent variable was the number of modifiable dementia factors (i.e., a continuous variable) so multiple linear regression as chosen. Additional information has been included in part 2.5.
- Justify the use of logistic regression analysis? What is the reason for that?
Response: Please see earlier response regarding the categorical nature of the dependent variable for the logistic regression.
- The results obtained should be discussed in a more professional manner.
Response: We encourage this reviewer to provide more concrete suggestions on how our results could be reported. We have followed the APA format for reporting for descriptive results as well as the regressions.
- Explain the theoretical implications of the study and reconsider the practical implications.
Response: We have added information about related theoretical constructs and practical implications in the discussion and conclusion including the Health Ageing model for holistic ageing and the Health Belief Model for understanding motivation to change behaviour.
- Where are the determinants of future study and research?
Response: We have provided some suggestions for implications and future directions for research in the discussion.
Reviewer 3 Report
I would like to suggest the authors to use more inferential statistics to establish some cause and effect relationship and establish the relationship in some variables and findings. From methodological point of view, the work is more descriptve in nature and except chi square.
Author Response
I would like to suggest the authors to use more inferential statistics to establish some cause and effect relationship and establish the relationship in some variables and findings. From methodological point of view, the work is more descriptve in nature and except chi square.
Response: We are unclear on this reviewer’s comment. The cross-sectional nature of this study is unable to establish cause and effect relationships between the variables of interest however we have conducted logistic and linear regression analyses to further estimate the relationship between the demographic variables with the dependent variable. We have attempted to clarify this further in the methods and results.
Reviewer 4 Report
Siette and colleagues in the present article entitled ‘Sociodemographic associations of dementia literacy in older Australians’, investigated whether dementia literacy has changed in older Australians. For this purpose, a 32-item adapted British Social Attitudes Survey was administered to 834 community-dwelling older adults on dementia awareness and knowledge of dementia risk and protective factors. Results showed that most of the respondents (61%) were aware of the relationship between different lifestyle factors and dementia risk, but very few were able to identify less well-known factors (e.g., chronic kidney disease); moreover, individuals with higher educational attainment were more likely to agree that dementia is modifiable. Authors concluded by stating a that tailored, evidence-informed, population-based lifespan approach targeting dementia literacy may help alleviate commonly reported barriers and support dementia risk reduction.
The main strength of this manuscript is that it addresses an interesting and timely question, providing a captivating interpretation and describing the importance of literacy in dementia awareness. In general, I think the idea of this research article is really interesting and the authors’ fascinating observations on this timely topic may be of interest to the readers of Journal of Ageing and Longevity. However, some comments, as well as some crucial evidence that should be included to support the authors’ argumentation, needed to be addressed to improve the quality of the manuscript, its adequacy, and its readability prior to its publication in the present form, in particular reshaping parts of the Introduction and Methods sections by adding more evidence.
Please consider the following comments:
· Abstract: According to the Journal’s guidelines, the abstract should be a total of about 200 words maximum. Please correct the actual one.
· Introduction: The ‘Introduction’ section is well-written and nicely presented, with a good balance of descriptive text and information about etiology and sympthomatology of dementia. Nevertheless, I believe that more information about pathophysiology of dementia, specifically of Alzheimer’s disease, its causes, symptoms and related neurocognitive changes, would provide a more defined and appropriate theoretical background. Considering that this study's main focus is to deepen current understanding of Alzheimer's treatment options, I would recommend citing a recent review that examined pathophysiological basis and biomarkers of AD pathology (https://doi.org/10.3390/ijms21249338) and recent studies that focused on how impairments in recognition of emotions, dysfunction in attention and in motor control, reflecting cognitive decline and earlier onset of cognitive impairment, are common in patients with neurologic and degenerative diseases (https://doi.org/10.3390/biomedicines10030627; https://doi.org/10.3390/geriatrics6010033; https://doi.org/10.3389/fnbeh.2022.946263).
· Measures: Data about participants and information about clinical assessment for patients’ selection are not adequately explained. For this reason, I would ask the authors to specify inclusion criteria for patients involved in this study, like severity of disorder. Also, could the authors specify how did they estimate the exact number of participants? Did they use a power analysis?
· Results: In my opinion, this section is well organized, but it illustrates findings in an excessively broad way. Authors should provide better describe statistical information, rewriting this section more accurately and not only presenting data in summary tables, to ensure in-depth understanding of their findings.
· Discussion: In this final section, authors described the results of their study and their argumentation and captured the state of the art well; however, I would have liked to see some views on a way forward. I believe that the authors should make an effort, trying to explain the theoretical implication as well as the translational application of this paper, to adequately convey what they believe is the take-home message of their study. Discussion of theoretical and methodological avenues in need of refinement is necessary, as well as suggestions of a path forward in understanding the significance of educating the public and suggest the use of multi-level implementation actions and behaviors from clinicians, organizations and governments, to support dementia prevention and application of new possibilities of treatment for patients at risk of AD. In this regard, recent evidence suggests that the application of new methods in Alzheimer’s treatment, such as the Non-invasive brain stimulation techniques (NIBS), have shown promising results in humans (https://doi.org/10.1097/WCO.0000000000000669). Importantly, I recommend referring recent studies that revealed that the application of NIBS induces long-lasting effects, noninvasively modulating the cortical excitability, and modulates a variety of cognitive functions: for example, a recent review acknowledged the implementation of NIBS to modulate in general fear memories (https://doi.org/10.1016/j.neubiorev.2021.04.036). In addition to the previous mentioned literature, authors might also see these additional studies that have focused on the efficacy of NIBS and IBS (https://doi.org/10.3389/fpsyt.2018.00201; https://doi.org/10.3389/fnagi.2020.578339).
· In my opinion, I think the ‘Conclusions’ paragraph would benefit from some thoughtful as well as in-depth considerations by the authors, because as it stands, it is very descriptive but not enough theoretical as a discussion should be. Authors should make an effort, trying to explain the theoretical implication as well as the translational application of their research.
· Figures and Tables: Please change the scale of the vertical axis and use the same minimum/maximum scale value in all the graphs in all the figures and reorganize the graphs’ space, to provide a better understanding and a direct interpretation of the results. Also, provide a fully descriptive caption for each table within the text.
· References: Authors should consider revising the bibliography, as there are several incorrect citations. Indeed, according to the Journal’s guidelines, they should provide the abbreviated journal name in italics, the year of publication in bold, the volume number in italics for all the references.
Overall, the manuscript contains 2 figures, 5 tables and 56 references. In my opinion, the manuscript might carry important value in describing the importance of literacy in dementia awareness.
I hope that, after these careful revisions, the manuscript can meet the Journal’s high standards for publication. I am available for a new round of revision of this article.
I declare no conflict of interest regarding this manuscript.
Best regards,
Reviewer
Author Response
Thank you for the detailed feedback - we feel these are quite critical in improving the message of our manuscript and hope we have addressed these satisfactorily.
Please consider the following comments:
- Abstract: According to the Journal’s guidelines, the abstract should be a total of about 200 words maximum. Please correct the actual one.
Response: The abstract has been edited, it is now 197 words.
- Introduction: The ‘Introduction’ section is well-written and nicely presented, with a good balance of descriptive text and information about etiology and sympthomatology of dementia. Nevertheless, I believe that more information about pathophysiology of dementia, specifically of Alzheimer’s disease, its causes, symptoms and related neurocognitive changes, would provide a more defined and appropriate theoretical background. Considering that this study's main focus is to deepen current understanding of Alzheimer's treatment options, I would recommend citing a recent review that examined pathophysiological basis and biomarkers of AD pathology (https://doi.org/10.3390/ijms21249338) and recent studies that focused on how impairments in recognition of emotions, dysfunction in attention and in motor control, reflecting cognitive decline and earlier onset of cognitive impairment, are common in patients with neurologic and degenerative diseases (https://doi.org/10.3390/biomedicines10030627; https://doi.org/10.3390/geriatrics6010033; https://doi.org/10.3389/fnbeh.2022.946263).
Response: Information about the early pathophysiological basis and biomarkers for dementia have been added to the introduction, lines 30-41.
- Measures: Data about participants and information about clinical assessment for patients’ selection are not adequately explained. For this reason, I would ask the authors to specify inclusion criteria for patients involved in this study, like severity of disorder. Also, could the authors specify how did they estimate the exact number of participants? Did they use a power analysis?
Response: We have included additional details regarding inclusion criteria although invite the reviewer to re-read 2.2 Study population and recruitment regarding the eligibility criteria for respondents. We have also now included additional sample size analysis details to justify sample size.
- Results: In my opinion, this section is well organized, but it illustrates findings in an excessively broad way. Authors should provide better describe statistical information, rewriting this section more accurately and not only presenting data in summary tables, to ensure in-depth understanding of their findings.
Response: We are unclear on how this section could be rewritten so that it is more accurate – we report sample characteristics, followed by a summary for each of the three main variables of interest namely (1) dementia literacy (descriptive, then inferential with logistic regression); (2) dementia awareness (descriptive, then inferential with multiple linear regression); and (3) barriers towards dementia risk reduction (descriptive only). Our manner of reporting was consistent with APA style and formatting but we are keen to know if there are alternative ways to summarise the data.
- Discussion: In this final section, authors described the results of their study and their argumentation and captured the state of the art well; however, I would have liked to see some views on a way forward. I believe that the authors should make an effort, trying to explain the theoretical implication as well as the translational application of this paper, to adequately convey what they believe is the take-home message of their study. Discussion of theoretical and methodological avenues in need of refinement is necessary, as well as suggestions of a path forward in understanding the significance of educating the public and suggest the use of multi-level implementation actions and behaviors from clinicians, organizations and governments, to support dementia prevention and application of new possibilities of treatment for patients at risk of AD. In this regard, recent evidence suggests that the application of new methods in Alzheimer’s treatment, such as the Non-invasive brain stimulation techniques (NIBS), have shown promising results in humans (https://doi.org/10.1097/WCO.0000000000000669). Importantly, I recommend referring recent studies that revealed that the application of NIBS induces long-lasting effects, noninvasively modulating the cortical excitability, and modulates a variety of cognitive functions: for example, a recent review acknowledged the implementation of NIBS to modulate in general fear memories (https://doi.org/10.1016/j.neubiorev.2021.04.036). In addition to the previous mentioned literature, authors might also see these additional studies that have focused on the efficacy of NIBS and IBS (https://doi.org/10.3389/fpsyt.2018.00201; https://doi.org/10.3389/fnagi.2020.578339).
Response: These papers as well as discussion of the potential of brain stimulation techniques and how they may work to complement lifestyle risk factors as treatments for Alzheimer’s have now been added to the Implications section.
- In my opinion, I think the ‘Conclusions’ paragraph would benefit from some thoughtful as well as in-depth considerations by the authors, because as it stands, it is very descriptive but not enough theoretical as a discussion should be. Authors should make an effort, trying to explain the theoretical implication as well as the translational application of their research.
Response: The Healthy Ageing model has been suggested as a model to guide dementia risk reduction interventions to holistically understand the ageing experience and align preventive action with individual and environmental resources available to communities. Please see Conclusions section
- Figures and Tables: Please change the scale of the vertical axis and use the same minimum/maximum scale value in all the graphs in all the figures and reorganize the graphs’ space, to provide a better understanding and a direct interpretation of the results. Also, provide a fully descriptive caption for each table within the text.
Response: The figures have now been relabelled as Figure 1,2,3 and 4 and the captions are included within the text to aid clarity. Using the same minimum and maximum value for the vertical axis of each of the graphs will make some of them particularly hard to read, especially due to the difference between figure 1 and figure 2.
- References: Authors should consider revising the bibliography, as there are several incorrect citations. Indeed, according to the Journal’s guidelines, they should provide the abbreviated journal name in italics, the year of publication in bold, the volume number in italics for all the references.
Response: References have been updated to suit the journal formatting
Reviewer 5 Report
The authors should differentiate between the general concept of dementia and the existing types of dementia, for example in line 34 they start talking about Alzheimer's after making reference to dementia figures in general, and there is some confusion
Author Response
The authors should differentiate between the general concept of dementia and the existing types of dementia, for example in line 34 they start talking about Alzheimer's after making reference to dementia figures in general, and there is some confusion.
Response: This sentence has now been removed to avoid confusion. Additional sentences have been added in lines 27-41 to generically describe dementia etiology, symptomatology, common types and it’s pathophysiological basis. This should also aid in clarifying how Alzheimer’s disease relates to dementia.
Round 2
Reviewer 1 Report
Thank you for addressing my suggestions and comments. I believe the manuscript has now reached the scientific merit and worthy of publication.